# Predicting Severe Haematological Toxicity in Gastrointestinal Cancer Patients Undergoing 5-FU-Based Chemotherapy: A Bayesian Network Approach

**DOI:** 10.3390/cancers15174206

**Published:** 2023-08-22

**Authors:** Oskitz Ruiz Sarrias, Cristina Gónzalez Deza, Javier Rodríguez Rodríguez, Olast Arrizibita Iriarte, Angel Vizcay Atienza, Teresa Zumárraga Lizundia, Onintza Sayar Beristain, Azucena Aldaz Pastor

**Affiliations:** 1Department of Mathematics and Statistic, NNBi, 31191 Esquiroz, Navarra, Spain; oskitz.ruiz@nnbi.es (O.R.S.);; 2Department of Medical Oncology, Clínica Universidad De Navarra, 31008 Pamplona, Navarra, Spain; cgdeza@unav.es (C.G.D.); jrodriguez@unav.es (J.R.R.); tzumarraga@unav.es (T.Z.L.); 3Pharmacy Unit, Clinica Universidad De Navarra, 31008 Pamplona, Navarra, Spain

**Keywords:** haematotoxicity prediction, 5-FU, neutropenia, thrombocytopenia, leukopenia, gastrointestinal cancer, Bayesian network, machine learning, artificial intelligence

## Abstract

**Simple Summary:**

Cancer treatments often have side effects that may impair patients’ quality of life. Our research aimed to create a predictive tool able to foresee the likelihood of these severe complications. We used medical data from 267 gastrointestinal cancer patients, applied a particular type of computer model known as a Bayesian network, and evaluated its predictions against real outcomes. The model accurately predicted the risk of developing severe haematological toxicity in 80–85% of cases. This tool, if further validated and refined, may help to identify a vulnerable subset of patients who might benefit from personalized treatment plans.

**Abstract:**

Purpose: Severe toxicity is reported in about 30% of gastrointestinal cancer patients receiving 5-Fluorouracil (5-FU)-based chemotherapy. To date, limited tools exist to identify at risk patients in this setting. The objective of this study was to address this need by designing a predictive model using a Bayesian network, a probabilistic graphical model offering robust, explainable predictions. Methods: We utilized a dataset of 267 gastrointestinal cancer patients, conducting preprocessing, and splitting it into TRAIN and TEST sets (80%:20% ratio). The RandomForest algorithm assessed variable importance based on MeanDecreaseGini coefficient. The bnlearn R library helped design a Bayesian network model using a 10-fold cross-validation on the TRAIN set and the aic-cg method for network structure optimization. The model’s performance was gauged based on accuracy, sensitivity, and specificity, using cross-validation on the TRAIN set and independent validation on the TEST set. Results: The model demonstrated satisfactory performance with an average accuracy of 0.85 (±0.05) and 0.80 on TRAIN and TEST datasets, respectively. The sensitivity and specificity were 0.82 (±0.14) and 0.87 (±0.07) for the TRAIN dataset, and 0.71 and 0.83 for the TEST dataset, respectively. A user-friendly tool was developed for clinical implementation. Conclusions: Despite several limitations, our Bayesian network model demonstrated a high level of accuracy in predicting the risk of developing severe haematological toxicity in gastrointestinal cancer patients receiving 5-FU-based chemotherapy. Future research should aim at model validation in larger cohorts of patients and different clinical settings.

## 1. Introduction

Most gastrointestinal cancer patients receive systemic therapy based on 5-FU during their treatment. Up to 30% of them will develop severe chemotherapy-related toxicity, a complication that can result in death for up to 1% of patients [1,2] and may lead to unnecessary hospital admissions, reduced quality of life, and negative impacts on overall survival [3,4]. The toxicity profile primarily includes myelosuppression, mucositis, nausea, vomiting, fatigue, and diarrhoea. Neurotoxicity and cardiotoxicity have also been reported [5]. The combination with other drugs, such as irinotecan, oxaliplatin, or docetaxel within approved conventional combinations, significantly worsens the toxicity profile [6,7,8].

Despite this high prevalence, few studies have been conducted to investigate the association between baseline clinical and analytical characteristics and the risk of developing severe toxicity [1,9,10,11,12,13,14]. As a result, current predictive tools for identifying at-risk patients are limited and lack personalization. In recent years, there has been growing interest in the use of machine learning and artificial intelligence techniques in oncology [15,16,17,18,19]. Bayesian networks have shown promise as they can effectively model complex relationships between multiple variables, incorporating prior knowledge and updating the model as new data become available [20,21,22,23,24]. These networks can provide a comprehensive understanding of the various factors contributing to the development of severe toxicities in patients undergoing 5-FU-based chemotherapy. This approach distinguishes our work from most existing predictive models, which often rely on black-box AI models [25].

The novelty of our study lies in the transparency and interpretability of our predictive model. Unlike black-box models such as deep learning, which are often criticized for their lack of interpretability [26], our Bayesian network model allows for a clear understanding of the relationships between variables. This transparency is crucial in the medical field, where understanding the reasoning behind a prediction can be as important as the prediction itself [27].

Given this context, our study aims to develop and validate a Bayesian network model for predicting severe toxicity in patients with gastrointestinal cancers receiving 5-FU-based chemotherapy. This approach has the potential to improve patient outcomes and healthcare efficiency, and to open new avenues for future research and model refinement [28].

We discuss the methodology used for constructing the network, including the selection of relevant variables and parameters, as well as the evaluation of the model’s performance in both training and validation sets. Finally, we outline the potential clinical implications of this approach and identify areas for future research and model refinement.

## 2. Methods

### 2.1. Patients

We conducted a retrospective analysis of 267 fit patients with locally advanced or metastatic gastrointestinal tumors, with adequate haematological, kidney, and liver functions who underwent systemic 5-FU-based chemotherapy, either as neoadjuvant or in the first-line setting, after evaluation by a multidisciplinary tumor board at our institution between 2010 and 2020.

Eligible patients were aged 18 years or older, had an Eastern Cooperative Oncology Group (ECOG) performance status of 0–1, and a confirmed diagnosis of either a locally advanced cancer (gastric, colon, rectal, or pancreatic) or liver metastasis from colorectal or pancreatic adenocarcinoma, all of them eligible for active systemic therapy. No patient received concurrent chemoradiation. To avoid potential bias, patients receiving second or further lines of treatment for metastatic disease were not included in the analysis.

### 2.2. Clinical Information

The variables retrospectively analyzed for each patient included gender, age, performance status, body mass index (BMI), histology, primary tumor location, clinical stage (cTNM), type of chemotherapy (FOLFOX, FOLFOXIRI, FLOT), endogenous pyrimidine levels, area under the curve of 5-fluorouracil, baseline analytical parameters (complete blood count, kidney and liver function, serum lactate dehydrogenase level), and toxicity profile.

Clinical toxicity: Patients were evaluated clinically for acute toxicity before starting chemotherapy, weekly during the first month and every two weeks thereafter. Toxicities were classified according to the National Cancer Institute Common Terminology Criteria for Adverse Events (CTCAE v5.0). Grade ≥ 3 toxicity was considered severe toxicity. A complete blood count and differential, urea, creatinine, and liver function tests (bilirubin, AST, ALT, Albumin, LDH) were performed before every cycle of treatment. Haematological toxicity was thus recorded every two weeks, given that all chemotherapy schedules analyzed were administered on a biweekly basis.

For the pharmacokinetic analyses of 5-FU, blood samples were taken at 15 and 30 h after start 5-FU infusion. 5-FU plasma concentrations were analyzed by high performance liquid chromatography (HPLC), according to the technique developed by the Clinical Pharmacokinetics Unit from our Center. The individual pharmacokinetic parameters of 5-FU were estimated by Bayesian methodology using, as preliminary information, a linear monocompartmental population model developed by the Clinical Pharmacokinetics Unit in 2002 and implemented in the USC* PACKv11.2 program package of the University of Southern California. The model was developed and validated in patients with colorectal and biliopancreatic cancer. Initially, both linear and non-linear pharmacokinetic models were explored, without finding significant differences in the selection criteria of the models. Therefore, similar to other models described, a linear monocompartmental model was selected for the individual estimation of 5-FU pharmacokinetic parameters [29,30].

### 2.3. Model Development

All the data processing, analysis, and development of the models outlined in this section were performed using the R statistical software (R version 4.2.2 2022-10-31) [31]. R is a widely used open-source programming language and software environment for statistical computing and graphics. The choice of R allowed us to efficiently perform data cleaning, preprocessing, and analysis, as well as develop the predictive models, leveraging its extensive range of libraries and functionalities tailored for statistical applications and predictive modeling.

### 2.4. Data Processing

The following steps were carried out to create a single, clean database containing individual patient information:Consolidation of databases: We combined individual patient databases into a single database, compiling all pertinent information for each patient.Standardization of variable formats: We ensured that all variables were standardized in terms of format, facilitating further analysis and processing.Removal of variables with excessive missing values: We excluded variables with missing values exceeding 10%. Those variables underwent thorough examination to ensure that their removal would not introduce bias or compromise the predictive capability of our model. This threshold was selected to strike a balance between retaining important information and ensuring the reliability of the model. Prior to their elimination, we verified the lack of a significant correlation with our target variable in order to preserve the integrity of our model.Imputation of missing values: For the remaining variables with missing values, we imputed the median value of the respective variable. This step allowed for the preservation of the overall structure and relationships within the data while accounting for missing information.Conversion of date variables into numerical variables: We transformed date variables into numerical variables by measuring the time elapsed between events of interest. This step enabled the incorporation of time-related information into the model.Identification of the target variable (Toxicity): We selected relevant toxicities for the study and recategorized the target variable as binary, where 1 represents the presence of severe toxicity of interest, and 0 represents its absence.Splitting the dataset into training and testing sets: We divided the cleaned database into a training set (80%) and a testing set (20%) to create and train the predictive model using the training set and perform an independent validation using the testing set. This approach allowed us to assess the model’s performance on unseen data and ensure its generalizability to new cases. Furthermore, the division of the datasets was performed in such a way that each set maintained the same proportion of the target variable, toxicity, ensuring a balanced distribution for a more accurate analysis.

### 2.5. Importance of Variables

In order to determine the importance of variables, we employed a robust approach that involved multiple iterations to minimize the variability in the MeanDecreaseGini coefficient associated with each variable [32,33]. This was achieved through the following procedure, which was executed 100 times:A random forest [34] model was constructed using 1000 decision trees, with toxicity as the target variable and all available variables serving as predictors.The MeanDecreaseGini coefficient for each variable was recorded, reflecting their importance in the model.

Upon completion of the 100 iterations, the mean MeanDecreaseGini coefficient for each variable was calculated across all iterations. This method was chosen to account for the inherent probabilistic nature of the RandomForest algorithm and to ensure a more reliable assessment of the importance of each variable. The results provided a ranking of the importance of all available variables, which was subsequently utilized in the construction of the Bayesian network.

### 2.6. Bayesian Network Model Design

In designing an optimal Bayesian network model, we employed the bnlearn R library, which specializes in Bayesian network design. The following steps were implemented:Database Augmentation: Due to the imbalanced nature of the original dataset, with more cases of individuals without toxicity compared with those with toxicity, we decided to augment the TRAIN dataset using the SMOTE (Synthetic Minority Over-sampling Technique) function from the performanceEstimation library [35,36] This approach aimed to balance the dataset by generating synthetic samples for the under-represented class, thereby enhancing the model’s ability to accurately predict severe toxicity.Dataset Partitioning: The TRAIN dataset was partitioned into 10 subgroups to facilitate the application of a 10-fold cross-validation scheme. This partitioning maintained a representative distribution of the target variable (toxicity) categories in each subgroup comparable to the overall TRAIN dataset.Optimization Strategy: In the analysis of mixed Bayesian networks using the bnlearn R library [37], the aic-cg method was employed for configuring the network structure. This method utilizes the Akaike Information Criterion (AIC) score [38,39] to select the optimal network structure, considering both numerical and categorical variables. By utilizing the aic-cg method, it is possible to obtain a network structure that balances model complexity and goodness of fit, leading to more accurate predictions and insights in the analysis of mixed data.
Adhering to the variable order determined by the previously obtained importance ranking, one variable at a time was incorporated into the model’s variable set, which initially contained only the target variable, toxicity.Cross-validation was performed on the updated variable set. In each iteration, a network structure was designed and its parameters determined using K-1 groups, while the remaining K group was employed to assess the predictive capacity of the preceding model.The cross-validation yielded a list of 10 estimations of the model’s predictive capacity. If the inclusion of a variable resulted in a higher set of estimations compared with the current model’s estimations, the variable’s incorporation was deemed successful.The iterative process continued until all variables were evaluated.
Model Validation: The predictive capability of the Bayesian network structure, which was established in the previous phase, was evaluated using both the TRAIN dataset for training and the TEST dataset for validation.

## 3. Results

### 3.1. Data Processing

Upon combining the separate patient databases into a unified, comprehensive database, we obtained a dataset consisting of 267 patients and a total of 195 variables. Many of these variables had only a limited number of measurements as they were analytical variables measured exclusively in small subsets of patients. This consolidation process facilitated efficient analysis and processing of the data in the subsequent stages of our study.

Out of the 195 variables, we excluded 155 variables with more than 10% of missing values to enhance the model’s reliability. For the remaining variables, we utilized median imputation to deal with missing values while retaining the data structure and relationships, resulting in a more comprehensive and representative dataset for further analysis and model development. Table 1 displays the variables that necessitated imputation, as well as the number of imputed values for each variable.

In order to incorporate time-related information into the model, we replaced the original variables with three new numerical variables: the difference in days between the day of the AUC1 measurement (Area under the curve of the first cycle) and the day of the first analytical measurement, the difference in days between the day of the AUC2 measurement and the day of the first analytical measurement, and the difference in days between the day of the AUC2 measurement and the day of the AUC1 measurement. This transformation ensured that the model could accurately account for the temporal relationships between these variables.

After performing all the necessary data preprocessing steps, a statistical summary of the variables that were used to build up the model is shown in Table 2 and Table 3.

In this study, we focused on severe toxicities categorized as grade 3 or 4, which were our target variable. As demonstrated in Table 4, the initial database showed a high level of variability in toxicities.

In response to this situation, we recategorized the toxicity variable as follows: 0 if the patient did not exhibit either neutropenia, thrombocytopenia or leukopenia; and 1 if they did. This decision is based on several factors. First, the prevalence of these toxicities in our patient population is significant, especially in the case of neutropenia, indicating that they are common reactions to the treatments used. Second, the severity of these conditions, all classified as grade 3 or 4 toxicities, can have a substantial impact on patient quality of life, making them critical variables for our study. Lastly, these types of toxicities are recognized as common side effects of chemotherapy treatments, underlining their relevance in our analysis.

### 3.2. Importance of Variables

The importance of variables in predicting severe toxicity was assessed using the MeanDecreaseGini coefficient obtained through the RandomForest algorithm, as described in the Section 2. The procedure was executed 100 times, each time constructing a random forest model with 1000 decision trees. Upon completion of the iterations, the mean MeanDecreaseGini coefficient for each variable was calculated across all iterations, providing a ranking of their importance in predicting severe toxicity. This ranking was instrumental in the construction of the Bayesian network.

To clarify, the MeanDecreaseGini coefficient is a metric commonly used in machine learning and specifically with random forests to determine the importance of variables in the model. It is computed by measuring how much the homogeneity of nodes and leaves—represented by Gini impurity, a measure of the likelihood an element is misclassified—decreases when they are split according to a particular variable. If a variable generates large decreases in Gini impurity, it is interpreted as highly important because it implies that the variable is more effective at separating cases into their correct classes.

The results of the variable importance analysis are presented in Table 5. These findings offer insight into the most significant factors influencing the development of grade 3–4 haematological toxicity in patients with gastrointestinal tumors treated with 5-FU-based chemotherapy.

### 3.3. Bayesian Network Model Design

#### Bayesian Network Structure

The optimal Bayesian network structure was determined through an iterative process using the aic-cg method. This structure represents the relationships among the variables, with the target variable, toxicity, at the center. A visual representation of the resulting Bayesian network is provided in Figure 1.

The Bayesian network structure provides a graphical representation of the relationships between the variables. The network’s structure can be summarized by the arcs connecting the nodes, as shown below:Origin is connected to: Dif Days AUC2 AUC1, Chemotherapy, CBC MCHC, Toxicity, AUC 1 and AUC 2.Toxicity is connected to: Dif Days AUC2 AUC1, AUC 1, and AUC 2.Age at diagnosis is connected to: CBC MCHC, AUC 2, and Dif Days AUC2 AUC1.AUC 1 is connected to: AUC 2 and CBC MCHC.AUC 2 is connected to: Dif Days AUC2 AUC1.CBC MCHC is connected to: Dif Days AUC2 AUC1.Chemotherapy is connected to: Toxicity.

The Bayesian network highlights the interconnectedness of the variables with the target variable, grade 3–4 haematological toxicity. Some variables, such as Origin of the tumor, have multiple connections, indicating their potential influence on multiple aspects of the prediction model. The network structure provides a visual representation of the dependencies and relationships among the variables, which can be further analyzed to gain insights into the factors that contribute to the prediction of severe toxicity in gastrointestinal cancer patients treated with 5-FU-based chemotherapy regimens.

### 3.4. Model Performance Metrics

The performance of the Bayesian network model was evaluated based on its accuracy, sensitivity, and specificity in predicting severe toxicity. The model was trained using 10-fold cross-validation on the TRAIN dataset and validated on the TEST dataset. The following metrics were obtained:

#### 3.4.1. Cross-Validation on TRAIN Dataset

Accuracy: The average accuracy of the model, obtained from the 10-fold cross-validation, indicates the proportion of correct predictions out of the total number of predictions made. In our study, the Bayesian network model achieved an average accuracy of 0.85 with a standard deviation of 0.05.Sensitivity: The average sensitivity measures the model’s ability to correctly identify patients who experience severe toxicity. Our model demonstrated an average sensitivity of 0.82 with a standard deviation of 0.14.Specificity: The average specificity assesses the model’s ability to correctly identify patients who do not experience severe toxicity. In this study, the Bayesian network model displayed an average specificity of 0.87 with a standard deviation of 0.07.

#### 3.4.2. Validation on TEST Dataset

Accuracy: The accuracy of the model on the TEST dataset was 0.80.Sensitivity: The sensitivity of the model on the TEST dataset was 0.71.Specificity: The specificity of the model on the TEST dataset was 0.83.

### 3.5. Model Implementation

Finally, a clinical tool that enables clinicians to visualize how variables within the model influence one another, offering valuable insight into the interrelationships be-tween variables contributing to severe toxicity prediction, has been designed.

The clinical tool, visualized in Figure 2 and Figure 3, serves a dual purpose in the context of severe toxicity prediction. Firstly, it facilitates the practical use of the Bayesian network model by clinicians in their predictive work. The left-hand sidebar of the tool’s interface allows clinicians to input measured values of the variables that constitute the Bayesian network. As additional variables are incorporated, the probability functions of the remaining variables undergo a dynamic transformation, redefining themselves in alignment with the probabilistic and interconnected nature of the Bayesian network. This nuanced interplay of variables can be explicitly observed in the transition from Figure 2 to Figure 3.

Secondly, the tool enables an empirical exploration of the Bayesian network’s internal operations. By manipulating input variables and observing the corresponding alterations in probabilities, clinicians gain a hands-on understanding of variable interconnections and can infer causative relations. This empirical investigation aids in further refining and enhancing the Bayesian network model, thereby optimizing its predictive accuracy for severe toxicity events.

With this two-fold functionality, the tool not only enhances the practical utility of the Bayesian network model but also presents an intuitive platform for clinicians to study and improve the model itself. This further solidifies its position as an effective and valuable instrument in predicting severe toxicity events, with the potential to contribute significantly to patient care.

## 4. Discussion

Defining the parameters that predict chemotherapy-induced toxicity is crucial to optimizing patient care. 5-Fluorouracil (5 FU) has a relatively narrow therapeutic window and wide inter-patient variability in toxicity. Not surprisingly, patients experience a wide range of toxicities and severities, with the primary dose-limiting complications being neutropenia, leukopenia, mucositis and diarrhoea [40,41].

Some of this variability in toxicity is accounted for by variations in drug pharmacokinetics (PK) [30], differences in co-morbidity (cardiac, liver, or renal diseases), predisposing factors (gender and age), variability in dihydropyrimidine dehydrogenase (DPD), activity of peripheral blood mononuclear cells (PBMNC), telomere length (TL) or platelet lymphocyte ratio (PLR) [42].

The currently available methods to identify patients at risk of developing 5-FU-related toxicity include pharmacogenetic assays to predict DPD activity [11,12,13], polymorphisms in Thymidylate synthase and Methylenetetrahydrofolate reductase [43], plasma measurement of uracil and the dihydrouracil ratio [8], determination of FUDR as an individual parameter of 5-FU degradation [44], fecal microbiota [28], or genetic variability in ABCC1/MRP1 genes [13]. However, none of these approaches include the whole spectrum of functional polymorphisms or do not include clinically relevant parameters, with the result that a considerable proportion of toxicity cases remain unexplained.

More recently, Wiberg et al. [45] employed machine learning techniques to predict neutropenic events using electronic medical records (EMRs). Their model, emphasizing interpretability and clinical applicability, achieved a commendable out-of-sample area under the receiver operating characteristic curve of 0.865 based on 20 clinical features. While their approach focused on leveraging EMRs and validated known risk factors, our study uses a Bayesian network model that offers a probabilistic perspective, allowing for a more nuanced understanding of the interplay between various factors. Moreover, our model’s emphasis on gastrointestinal cancer and the specific chemotherapy regimens provides a specialized lens, potentially offering more targeted predictions for this subset of patients.

The aim of our study was to develop a model that could predict the risk of developing severe haematological toxicity in patients undergoing 5-FU-based chemotherapy regimens for gastrointestinal cancer. The resulting Bayesian network model demonstrated high accuracy in predicting severe toxicity related to these specific conditions, with an average accuracy of 0.85 in cross-validation on the TRAIN dataset and an accuracy of 0.80 on the TEST dataset. The model’s sensitivity and specificity were also satisfactory, indicating its ability to identify both patients who would and would not experience severe toxicity.

Our proposed model, while demonstrating high accuracy, does have several limitations that warrant further discussion. The retrospective nature of our study, for instance, may introduce potential biases such as selection bias and information bias. The data used were not collected with the specific intention of developing this model, which may have led to the omission of potentially relevant variables. Moreover, the historical records we relied on may not fully capture the complexity of the patient’s condition at the time of treatment, which could potentially impact the accuracy of our model when applied prospectively.

Another significant limitation is the sample size of our study. The number of patients included, while sufficient for the development of our initial model, may not be large enough to fully capture the variability in patient responses to 5-FU-based chemotherapy. This limited sample size could affect the robustness of our model and its ability to generalize to a broader population of patients. Future research should aim to validate and refine our model using larger datasets, enhancing the statistical power, and improving the model’s performance and generalizability.

The focus of our model on specific cancer types and treatment regimens could limit its generalizability. While the model performed well in predicting severe haematological toxicity in patients undergoing 5-FU-based chemotherapy for gastrointestinal cancer, its performance may not be as robust when applied to patients with different types of cancer or those undergoing different treatment regimens. Therefore, future studies should aim at validating and potentially adapting our model to a broader range of cancer types and treatment regimens.

Another limitation of our study is the potential impact of missing data and the imputation methods used. Missing data can introduce bias and reduce the statistical power of a study. In our case, we used median imputation to handle missing data. While this method is robust to outliers and can be a reasonable approach when data are missing at random, it assumes that the missing values are not significantly different from the observed ones. If this assumption is incorrect, median imputation could have introduced additional bias. Furthermore, the imputation of variables such as Blood creatinine and MDRD (GFR algorithm) using the median may have affected the model’s performance, as these variables are critical in assessing a patient’s renal function and potential toxicity response. The use of median values could potentially oversimplify the variability and distribution of these critical variables. Future studies should aim to minimize missing data and carefully consider the most appropriate imputation methods, potentially exploring multiple imputation techniques or model-based methods that might provide more accurate estimates.

Despite these limitations, our Bayesian network model demonstrates high accuracy in predicting severe haematological toxicity in gastrointestinal cancer patients undergoing a 5-FU-based chemotherapy regimen. It provides a clear representation of the relationships among variables and how they influence the prediction of severe toxicity related to neutropenia, leukopenia, and thrombocytopenia. This transparency contributes to the trustworthiness of the model in a clinical setting and enables a better understanding of its predictions. The model’s performance metrics suggest its potential usefulness in clinical practice for identifying patients at risk of severe toxicity and enabling more personalized treatment approaches to improve patient outcomes. However, the limitations discussed above should be considered when interpreting its findings and applying it in clinical practice. Future research should aim to address these limitations to improve the model’s generalizability and reliability.

## 5. Conclusions

Our study indicates that the Bayesian network model developed has potential for enhancing the predictability of severe toxicity, particularly neutropenia, leukopenia, and thrombocytopenia, in patients with gastrointestinal cancer undergoing 5-FU-based chemotherapy regimens. It is important to underscore, however, the limitations inherent to our study, most notably the limited sample size, which may have influenced the model’s performance.

The limited sample size not only potentially reduces the statistical power of our findings, but it may also limit the model’s ability to fully capture the intricate and complex interactions of variables within a larger, more diverse patient population. The nature of these interactions could vary substantially in a broader population, which may impact the effectiveness of the model’s predictions.

Moving forward, additional research should prioritize validating this model within larger, independent cohorts, and addressing the issues of missing data and sample size. The focus should also be on incorporating a broader range of variables into the model to further refine its accuracy and robustness, thereby enhancing its predictive power for severe toxicity among this patient demographic.

Future research endeavors could explore collaboration efforts to accumulate and analyze larger datasets. These initiatives would provide a more comprehensive understanding of variable interactions in a wider patient population and contribute to the refinement of the model, enhancing its reliability and applicability in predicting severe toxicity. Through such concerted efforts, we can work towards developing a more precise and robust model for predicting severe toxicity in patients undergoing 5-FU-based chemotherapy.

## Figures and Tables

**Figure 1 cancers-15-04206-f001:**
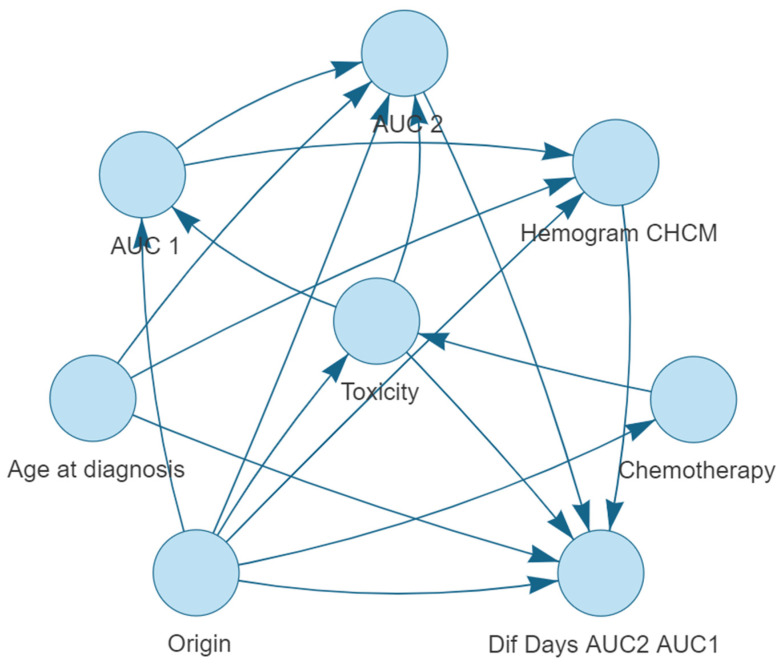
Bayesian network structure for predicting severe haematological toxicity (neutropenia, leukopenia and thrombocytopenia) in gastrointestinal cancer patients treated with 5-FU-based chemotherapy regimens.

**Figure 2 cancers-15-04206-f002:**
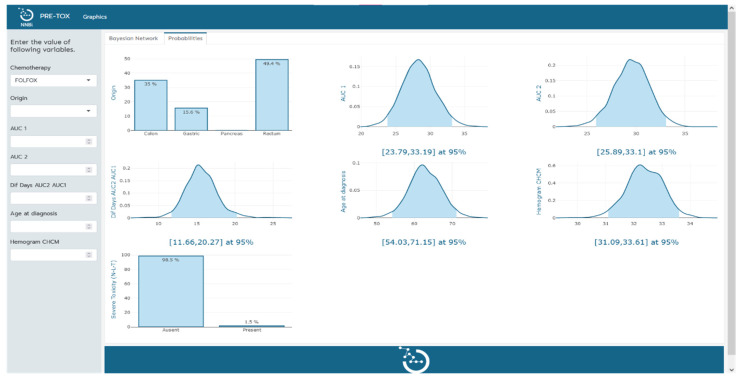
Tool implementation. Visual representation of the probabilities of the remaining variables in the model upon determination of the chemotherapy type.

**Figure 3 cancers-15-04206-f003:**
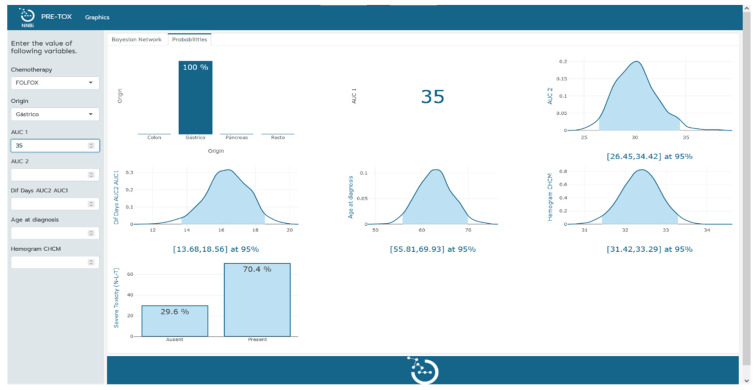
Tool implementation. Visual representation of the probabilities of the remaining variables in the model upon determination of chemotherapy type, origin, and AUC 1.

**Table 1 cancers-15-04206-t001:** Variables that required imputation, along with the number of imputed values for each variable.

Variable	Number of Imputed Values
Area under the curve of the second cycle (AUC_2)	2
CBC Platelet amplitude distribution (PDW)	4
CBC PTC	4
CBC Red Cell Distribution Width (RDW)	1
CBC Mean Platelet Volume (MPV)	1
Plasma creatinine	23
MDRD (GFR algorithm)	24

CBC = Cell Blood Count.

**Table 2 cancers-15-04206-t002:** Statistical summary of the numeric variables used for the construction of the model.

Variable	Min	1st.Qu.	Median	Mean	3rd.Qu.	Max	SD
Age at diagnosis	27.04	57.092	64.819	63.714	71.712	90.808	10.762
BMI	15.800	23.400	26.000	26.068	28.300	44.200	4.250
Dif_Days_AUC1_Analy1	−12.000	2.000	2.000	3.974	5.000	23.000	3.740
Dif_Days_AUC2_Analy1	2.000	16.000	18.000	21.401	22.000	381.000	25.574
Dif_Days_AUC2_AUC1	11.000	14.000	14.000	17.464	16.000	379.000	25.475
AUC-1	13.500	25.000	29.000	29.536	32.500	59.700	6.762
AUC-2	13.700	26.000	28.700	28.704	31.000	54.000	5.043
CBC Bas (10^9^/L)	0	0.020	0.040	0.043	0.050	0.190	0.026
CBC Basophils (%)	0	0.300	0.500	0.542	0.700	1.700	0.288
CBC MCHC (g/dL)	27.200	31.700	32.600	32.472	33.400	51.800	2.017
CBC Eos (10^9^/L)	0	0.080	0.130	0.192	0.235	2.260	0.244
CBC Eosinophils (10^9^/L)	0	1.100	1.800	2.324	3.100	21.000	2.242
CBC Hb (g/dL)	7.400	11.400	13.300	12.880	14.400	17.700	1.905
CBC HCM (pg)	17.600	27.400	29.300	28.855	30.700	52.500	3.548
CBC Hematies (10^12^/L)	2.440	4.145	4.520	4.480	4.870	6.070	0.55
CBC Hto (%)	24.200	35.900	40.500	39.625	43.350	53.900	5.233
CBC Leukocytes (10^9^/L)	3320	6.425	7.640	8.216	9.645	30.850	2.554
CBC Lin (10^9^/L)	0.460	1.395	1.750	1.792	2.085	4.080	0.621
CBC Lymphocytes (%)	2.700	17.650	23.500	23.420	28.250	49.500	8.234
CBC Mon (10^9^/L)	0.190	0.485	0.610	0.652	0.770	1.650	0.239
CBC Monocytes (%)	1.500	6.600	8.100	8.175	9.550	18.500	2.295
CBC Neu (10^9^/L)	1.360	3.965	5.050	5.531	6.545	28.150	2.208
CBC Neutrophils (%)	35.600	60.500	65.600	65.494	71.400	92.800	9.430
CBC PDW (fL)	8.700	11.500	12.900	13.889	15.250	66.700	4.157
CBC Platelet (10^9^/L)	85.000	207.000	267.000	278.798	343.000	595.000	96.363
CBC PTC (%)	0.080	0.200	0.280	0.281	0.350	0.640	0.094
CBC RDW (%)	11.600	12.850	13.600	14.564	15.150	30.300	2.928
CBC VCM (fL)	64.300	85.950	89.300	88.656	92.900	108.000	7.233
CBC VPM (fL)	6.500	9.500	10.200	10.193	10.950	13.500	1.225
Blood Creatinine (mg/dL)	0.400	0.700	0.800	0.872	1.000	2.900	0.234
MDRD(GFR)(mL/min/1.73 m^2^)	23.000	80.000	92.000	92.581	105.000	171.000	21.637

CBC = Cell Blood Count.

**Table 3 cancers-15-04206-t003:** Statistical summary of the categorical variables used for the construction of the model.

Variable	Category	Number
Sex	Female	79
	Male	188
ECOG	0	111
	1	156
Histology	Adenocarcinoma	263
	Carcinoma	4
Origin	Colon	81
	Gastric	45
	Pancreas	82
	Rectum	59
Stage	Disseminated	76
	Localized	21
	Regional	170
Type of	FLOT	18
Chemotherapy	FOLFOX	134
	FOLFOXIRI	115
Pyrimidines	Altered	11
Metabolism	Normal	256
Patient Status	Dead with disease	108
	Dead without disease	5
	Alive with disease	34
	Alive without disease	120

**Table 4 cancers-15-04206-t004:** Summary of grade 3–4 toxicities.

Toxicities	Patients
Anemia	6/267
Asthenia	6/267
Diarrhoea	11/267
Hyporexia	1/267
Leukopenia	8/267
Lymphopenia	13/267
Mucositis	3/267
Neutropenia	66/267
Nausea/vomiting	2/267
Rash	1/267
Neurological toxicity	1/267
Cardiac toxicity	1/267
Thrombocytopenia	5/267

**Table 5 cancers-15-04206-t005:** Importance of variables based on mean MeanDecreaseGini coefficient across 100 iterations.

Variable	MeanDecreaseGini
Dif_Days_AUC2_AUC1	7.68418463
Chemotherapy	6.63702847
AUC-1	5.98405905
Origin	3.66820382
Dif_Days_AUC2_Analy1	3.12617860
MDRD (GFR algorithm)	2.75232908
AUC-2	2.64474938
CBC Eos	2.51249340
CBC PDW	2.51066431
CBC Platelet	2.35070235
CBC Leukocytes	2.20782943
BMI	2.16055155
CBC RDW	2.11807334
Age at diagnosis	2.09749240
CBC Eosinophils	2.06942951
CBC Mon	2.04356005
CBC Neutrophils	2.03195574
CBC Hematies	1.99520591
CBC Neu	1.94319862
CBC Lymphocytes	1.92257386
CBC Lin	1.90860498
CBC Hb	1.79924777
CBC MCHC	1.75612465
CBC Hto	1.72569409
CBC PTC	1.67414576
CBC Monocytes	1.60481585
CBC VCM	1.58960997
CBC VPM	1.57007868
CBC HCM	1.53288421
Blood Creatinine	1.33964382
CBC Basophils	1.00624322
Dif_Days_AUC1_Analy1	0.95062264
CBC Bas	0.91029812
Status	0.80868056
Stage	0.54241400
ECOG	0.50190054
Sex	0.19311684
Pyrimidines *	0.05256130
Histology	0.02240322

CBC = Cell Blood Count, * Meulendijks Criteria.

## Data Availability

No new data were created or analyzed in this study. Data sharing is not applicable to this article.

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
