# Peer review of "Predicting Severe Haematological Toxicity in Gastrointestinal Cancer Patients Undergoing 5-FU-Based Chemotherapy: A Bayesian Network Approach"

_cancers, 2023, doi:10.3390/cancers15174206_

Round 1

Reviewer 1 Report

In the present manuscript, the Authors aimed at developing a predictive model for severe hematologic toxicity in GI cancer undergoing 5-FU based chemotherapy, using Bayesan network. The topic is of interest. Few comments:

1)     Toxicity evaluation. Can you please specify the timeframe in which toxicity has been accessed? Within 1 month from treatment start? Other?

2)     Are we focusing on acute toxicity in this study? Please specify.

3)     Please use Grade 3 instead of Grade III

4)     I assume the toxicity profile would depend also on the chemotherapy line considered, being secondo-third line therapies more likely to develop toxicity compared to first line therapies. Do you have this data? Can it be incorporated in the model. If not this should be acknowledged in the limitation.

5)     Part of the hematologic toxicity can be due to radiation therapy (rectal, pancreatic, gastric cancer. Do you have this data? Can it be incorporated in the analysis. If not this should be acknowledged in the limitation. Please cite: Franco P, Arcadipane F, Ragona R, et al. Hematologic toxicity in anal cancer patients during combined chemo-radiation: a radiation oncologist perspective. Expert Rev Anticancer Ther 2017;17:335-45.

6)     Data processing. I think this paragraph can be shortened. Some of the data quality analysis is standard procedure in statistical analysis.

Moderate revision of the English language is required

Author Response

Attached you will find a one by one response to all the suggestions elicited by the reviwers.

We´d like to thank the revisions sent along, as we feel that the manuscript`s quality has greatly improved due to these suggestions

We hope that the manuscript fulfils the standards of quality of the journal Cancers

Please let us know whether you need any other clarification

Kind regards

Oskitz Ruiz

1)     Toxicity evaluation. Can you please specify the timeframe in which toxicity has been accessed? Within 1 month from treatment start? Other? This important issue has now been clarified in lines 106-108.

2)     Are we focusing on acute toxicity in this study? Please specify. This has been clarified in line 101

3)     Please use Grade 3 instead of Grade III. Corrected throughout the manuscript

4)     I assume the toxicity profile would depend also on the chemotherapy line considered, being secondo-third line therapies more likely to develop toxicity compared to first line therapies. Do you have this data? Can it be incorporated in the model. If not this should be acknowledged in the limitation. No patient in this analysis received second or further lines of therapy. This has been specified in lines 91-93

5)     Part of the hematologic toxicity can be due to radiation therapy (rectal, pancreatic, gastric cancer. Do you have this data? Can it be incorporated in the analysis. If not this should be acknowledged in the limitation. Please cite: Franco P, Arcadipane F, Ragona R, et al. Hematologic toxicity in anal cancer patients during combined chemo-radiation: a radiation oncologist perspective. Expert Rev Anticancer Ther 2017;17:335-45.No patient received concurrent radiotherapy during toxicity evaluation. This has now been specified in line 91-93.

6)     Data processing. I think this paragraph can be shortened. Some of the data quality analysis is standard procedure in statistical analysis. We appreciate the reviewer's suggestion to shorten the data processing section. However, we decided to maintain its current length to ensure full transparency of the methodological approach. The level of detail presented in this section is intended to facilitate reproducibility of our results, which is a cornerstone of scientific research.

Reviewer 2 Report

The study focuses on predicting severe toxicity in gastrointestinal cancer patients undergoing 5-FU-based chemotherapy, aiming to optimize patient care.

Title:

Represent the study very well, concise 

Abstract:

  1. Acknowledging the restrictions in abstract length I believe that the abstract lacks a concise statement of the research gap or rationale for the study. It would be helpful to briefly mention why predicting the risk of severe hematological toxicity in gastrointestinal cancer patients receiving 5-Fluorouracil-based chemotherapy is important or relevant in clinical practice.
  2. The term "Bayesian networks" is mentioned without sufficient background or context. It would be beneficial to provide a brief explanation of what Bayesian networks are and how they are relevant to the study's objectives.

Introduction:

1.      The introduction provides a general overview of the prevalence of systemic therapy based on 5-FU in gastrointestinal cancer patients and the occurrence of chemotherapy-related severe toxicity. However, it lacks specific references to support the stated percentages (up to 30% developing severe toxicity and approximately 1% leading to death). Providing citations to relevant studies in the literature would strengthen the accuracy and scientific soundness of these claims.

2.      The toxicity profile mentioned aligns with common adverse effects associated with 5-FU-based chemotherapy, but it lacks specific references to support the statement. Including references to clinical studies or guidelines that outline the documented toxicity profile would enhance the credibility of the information presented.

3.      The statement that "few studies have investigated the association between baseline clinical and analytical characteristics and the risk of developing severe toxicity" lacks specific references or evidence to support this claim. Providing citations to existing studies that have explored the relationship between baseline characteristics and severe toxicity in gastrointestinal cancer patients receiving 5-FU-based chemotherapy would strengthen this assertion.

4.      The introduction lacks a clear statement of the research gap or rationale for the study. It would be helpful to explicitly mention why developing a Bayesian network model for predicting severe toxicity in gastrointestinal cancer patients receiving 5-FU-based chemotherapy is important or relevant in clinical practice. Providing a brief background on the limitations of current approaches or the need for personalized risk prediction tools would strengthen the introduction.

5.      The objective of developing and validating a Bayesian network model for predicting severe toxicity is clearly stated. However, it would be beneficial to provide more specific details on the novelty or unique contribution of this study compared to existing literature. Highlighting the specific variables or factors included in the model that have not been previously investigated would strengthen the significance of the study.

 Methods:

1.      The decision to exclude variables with more than 10% missing values may introduce bias if those variables are associated with the target variable or have significant predictive power. It would be beneficial to provide a rationale for this threshold and discuss any potential implications.

Results:

1.      The use of median imputation for handling missing values is a simplistic approach that may not accurately capture the true underlying patterns in the data. Considering more advanced imputation methods, such as multiple imputation or machine learning-based approaches, could improve the integrity of the dataset.

2.      The statistical summaries in Tables 2 and 3 lack important descriptive statistics such as standard deviations and interquartile ranges. Including these measures would provide a more complete understanding of the variable distributions and aid in assessing their variability.

3.      The MeanDecreaseGini coefficient obtained through the RandomForest algorithm is mentioned as the measure of variable importance, but no explanation is provided regarding its interpretation or significance. Elaborating on the meaning of this coefficient and its relevance to the study would help readers understand its implications better.

4.      The clinical tool presented in Figures 2 and 3 is described briefly without sufficient detail. Explaining the functionality and specific insights that can be gained from the tool would better convey its utility for clinicians.

Discussion:

1.      While the discussion highlights previous studies that have investigated predictors of severe toxicity in patients with digestive tumors, it could provide a more comprehensive overview of the existing literature. Including a discussion of the findings and limitations of these studies would help contextualize the current study and its contributions.

 2.      The mention of Wiberg et al.'s work on machine learning-based risk assessment is interesting but lacks a direct connection to the current study. Elaborating on the similarities and differences between their approach and the Bayesian network model developed in this study would strengthen the discussion and highlight the unique contributions of the current research.

 3.      The limitations of the proposed model are briefly mentioned, but further discussion and exploration of these limitations would be beneficial. For example, the retrospective nature of the study, the restricted focus on specific cancer types and treatment regimens, and the potential impact of missing data and imputation methods could be discussed in more detail, along with their potential implications for the model's generalizability and reliability.

 4.      The discussion acknowledges the limited sample size and its potential impact on the model's performance. However, it could further address the implications of the sample size limitations and suggest avenues for future research, such as the need for larger datasets to validate and refine the model's performance.

Conclusions:

The discussion acknowledges the limited sample size and its potential impact on the model's performance. However, it could be expanded to further address the implications of the sample size limitations. This could involve discussing the potential limitations in capturing the complexity of variable interactions in a larger patient population and the need for additional studies with larger datasets to validate and refine the model's performance. Suggestions for future research avenues, such as collaborative efforts to collect and analyze larger datasets, could also be provided.

Author Response

Attached you will find a one by one response to all the suggestions elicited by the reviwers.

We´d like to thank the in-depth revisions sent along, as we feel that the manuscript`s quality has greatly improved due to these suggestions

We hope that the manuscript fulfils the standards of quality of the journal Cancers

Please let us know whether you need any other clarification

Kind regards

Oskitz Ruiz

Title:

Represent the study very well, concise 

Abstract:

  1. Acknowledging the restrictions in abstract length I believe that the abstract lacks a concise statement of the research gap or rationale for the study. It would be helpful to briefly mention why predicting the risk of severe hematological toxicity in gastrointestinal cancer patients receiving 5-Fluorouracil-based chemotherapy is important or relevant in clinical practice. A brief mention of the relevance of predicting haematological toxicity has been added in lines 22-24
  2. The term "Bayesian networks" is mentioned without sufficient background or context. It would be beneficial to provide a brief explanation of what Bayesian networks are and how they are relevant to the study's objectives. A brief explanation of what Bayesian networks are has been included in line 25-26.

Introduction:

  1. The introduction provides a general overview of the prevalence of systemic therapy based on 5-FU in gastrointestinal cancer patients and the occurrence of chemotherapy-related severe toxicity. However, it lacks specific references to support the stated percentages (up to 30% developing severe toxicity and approximately 1% leading to death). Providing citations to relevant studies in the literature would strengthen the accuracy and scientific soundness of these claims. Several additional citations have been added as suggested by the reviewer.
  2. The toxicity profile mentioned aligns with common adverse effects associated with 5-FU-based chemotherapy, but it lacks specific references to support the statement. Including references to clinical studies or guidelines that outline the documented toxicity profile would enhance the credibility of the information presented. Clinical studies or guidelines that outline the documented toxicity profile have been now incorporated to enhance the credibility of the information presented.

  1. The statement that "few studies have investigated the association between baseline clinical and analytical characteristics and the risk of developing severe toxicity" lacks specific references or evidence to support this claim. Providing citations to existing studies that have explored the relationship between baseline characteristics and severe toxicity in gastrointestinal cancer patients receiving 5-FU-based chemotherapy would strengthen this assertion. As suggested, citations have been added regarding studies analysing the relationship between baseline features and toxicity.
  2. The introduction lacks a clear statement of the research gap or rationale for the study. It would be helpful to explicitly mention why developing a Bayesian network model for predicting severe toxicity in gastrointestinal cancer patients receiving 5-FU-based chemotherapy is important or relevant in clinical practice. Providing a brief background on the limitations of current approaches or the need for personalized risk prediction tools would strengthen the introduction. This has now been added in lines 54-56 and 59-65.
  3. The objective of developing and validating a Bayesian network model for predicting severe toxicity is clearly stated. However, it would be beneficial to provide more specific details on the novelty or unique contribution of this study compared to existing literature. Highlighting the specific variables or factors included in the model that have not been previously investigated would strengthen the significance of the study. More specific details on the novelty of this study are now available in lines 66-71.

 Methods:

  1. The decision to exclude variables with more than 10% missing values may introduce bias if those variables are associated with the target variable or have significant predictive power. It would be beneficial to provide a rationale for this threshold and discuss any potential implications. More specific details are now available in lines 137-143.

Results:

  1. The use of median imputation for handling missing values is a simplistic approach that may not accurately capture the true underlying patterns in the data. Considering more advanced imputation methods, such as multiple imputation or machine learning-based approaches, could improve the integrity of the dataset. This issue is now discussed in lines 229-236.
  2. The statistical summaries in Tables 2 and 3 lack important descriptive statistics such as standard deviations and interquartile ranges. Including these measures would provide a more complete understanding of the variable distributions and aid in assessing their variability. The required measures have been incorporated according to the reviewer suggestion.
  3. The MeanDecreaseGini coefficient obtained through the RandomForest algorithm is mentioned as the measure of variable importance, but no explanation is provided regarding its interpretation or significance. Elaborating on the meaning of this coefficient and its relevance to the study would help readers understand its implications better. The meaning of this coefficient and its relevance to the study is now detailed in lines 265-271.
  4. The clinical tool presented in Figures 2 and 3 is described briefly without sufficient detail. Explaining the functionality and specific insights that can be gained from the tool would better convey its utility for clinicians. Figures 2 and 3 are now explained in more detail in lines 332-350.

Discussion:

  1. While the discussion highlights previous studies that have investigated predictors of severe toxicity in patients with digestive tumors, it could provide a more comprehensive overview of the existing literature. Including a discussion of the findings and limitations of these studies would help contextualize the current study and its contributions. A more comprehensive overview of the existing literature is now provided in lines 368-374.
  2. The mention of Wiberg et al.'s work on machine learning-based risk assessment is interesting but lacks a direct connection to the current study. Elaborating on the similarities and differences between their approach and the Bayesian network model developed in this study would strengthen the discussion and highlight the unique contributions of the current research. Wiberg et al.'s work is further discussed in lines 376-384.
  3. The limitations of the proposed model are briefly mentioned, but further discussion and exploration of these limitations would be beneficial. For example, the retrospective nature of the study, the restricted focus on specific cancer types and treatment regimens, and the potential impact of missing data and imputation methods could be discussed in more detail, along with their potential implications for the model's generalizability and reliability. Further discussion and exploration of the limitations of the study can be found in lines 393-427.
  4. The discussion acknowledges the limited sample size and its potential impact on the model's performance. However, it could further address the implications of the sample size limitations and suggest avenues for future research, such as the need for larger datasets to validate and refine the model's performance. Some potential avenues of potential research have been highlighted in lines 458-464.

Conclusions:

The discussion acknowledges the limited sample size and its potential impact on the model's performance. However, it could be expanded to further address the implications of the sample size limitations. This could involve discussing the potential limitations in capturing the complexity of variable interactions in a larger patient population and the need for additional studies with larger datasets to validate and refine the model's performance. Suggestions for future research avenues, such as collaborative efforts to collect and analyze larger datasets, could also be provided. Some potential avenues of potential research have been highlighted in lines 458-464

Round 2

Reviewer 1 Report

No further comments.

Suitable quality fo english language

Reviewer 2 Report

Authors have done great job addressing my comments and concerns. I have no further comments.